# Implementing artificial intelligence in Canadian primary care: Barriers and strategies identified through a national deliberative dialogue

Katrina Darcel[1,2], Tara Upshaw[1,3], Amy Craig-Neil[1], Jillian Macklin[1,2,4,5], Carolyn Steele Gray[6,7], Timothy C. Y. Chan [8], Jennifer Gibson[4,5], Andrew D. Pinto [1,5,9,10]*

1 Upstream Lab, MAP Centre for Urban Health Solutions, Unity Health Toronto, Toronto, Ontario, Canada, 2 Undergraduate Medical Education, Temerty Faculty of Medicine, University of Toronto, Toronto, Ontario, Canada, 3 Cumming School of Medicine, University of Calgary, Calgary, Alberta, Canada, 4 Joint Centre for Bioethics, University of Toronto, Toronto, Ontario, Canada, 5 Dalla Lana School of Public Heath, University of Toronto, Toronto, Ontario, Canada, 6 Bridgepoint Collaboratory for Research and Innovation, Lunenfeld-Tanenbaum Research Institute, Sinai Health System, Toronto, Ontario, Canada, 7 Institute of Health Policy, Management and Evaluation, Dalla Lana School of Public Health, University of Toronto, Toronto, Ontario, Canada, 8 Department of Mechanical and Industrial Engineering, Faculty of Applied Science and Engineering, University of Toronto, Toronto, Ontario, Canada, 9 Department of Family and Community Medicine, St. Michael's Hospital, Toronto, Ontario, Canada, 10 Department of Family and Community Medicine, Faculty of Medicine, University of Toronto, Toronto, Ontario, Canada

* andrew.pinto@utoronto.ca

**Data Availability Statement:** All relevant data are in the paper and its Supporting Information files.

## Abstract

### Background

With large volumes of longitudinal data in electronic medical records from diverse patients, primary care is primed for disruption by artificial intelligence (AI) technology. With AI applications in primary care still at an early stage in Canada and most countries, there is a unique opportunity to engage key stakeholders in exploring how AI would be used and what implementation would look like.

### Objective

To identify the barriers that patients, providers, and health leaders perceive in relation to implementing AI in primary care and strategies to overcome them.

### Design

12 virtual deliberative dialogues. Dialogue data were thematically analyzed using a combination of rapid ethnographic assessment and interpretive description techniques.

### Setting

Virtual sessions.

**Funding:** This project was supported by the Canadian Institutes for Health Research (FRN 156885), with ADP as Principal Investigator. TLU's time was supported by a CIHR Frederick Banting and Charles Best Canada Graduate Scholarship. ADP is supported by the Department of Family and Community Medicine, Temerty Faculty of Medicine at the University of Toronto and at St. Michael's Hospital, and by the Li Ka Shing Knowledge Institute, Unity Health Toronto. He is also supported by a CIHR Applied Public Health Chair and a fellowship from the Physicians' Services Incorporated Foundation. The funders had no role in study design, data collection and analysis, decision to publish, or preparation of the manuscript.

**Competing interests:** The authors have declared that no competing interests exist.

## Participants

Participants from eight provinces in Canada, including 22 primary care service users, 21 interprofessional providers, and 5 health system leaders

## Results

The barriers that emerged from the deliberative dialogue sessions were grouped into four themes: (1) system and data readiness, (2) the potential for bias and inequity, (3) the regulation of AI and big data, and (4) the importance of people as technology enablers. Strategies to overcome the barriers in each of these themes were highlighted, where participatory co-design and iterative implementation were voiced most strongly by participants.

## Limitations

Only five health system leaders were included in the study and no self-identifying Indigenous people. This is a limitation as both groups may have provided unique perspectives to the study objective.

## Conclusions

These findings provide insight into the barriers and facilitators associated with implementing AI in primary care settings from different perspectives. This will be vital as decisions regarding the future of AI in this space is shaped.

## Introduction

With advancements in artificial intelligence (AI) and vast volumes of health data produced through electronic health records (EHRs), we are in an opportune time to explore ways in which AI can be applied in healthcare settings. Although many of the advancements have taken place in the field of radiology [1], AI applications such as online scheduling tools, drug dosing algorithms, and digitization of medical records have been implemented in many other areas of healthcare, including primary care [2].

Applying AI in primary care settings poses unique challenges as this field of medicine encompasses a wide variation of health concerns in diverse patient populations [3]. Primary care EHR data is unique in that it is longitudinal, it covers undifferentiated and broad health concerns, it serves populations of differing socioeconomic statuses, it contributes to public health records, and it contains data to inform preventative care practices. Despite these distinct qualities, AI is beginning to be used in primary care in the areas of clinical decision making, risk prediction, care management, and proactive detection [4]. Examples of risk prediction AI tools applicable to the primary care space include machine learning algorithms that identify patients at risk for cancer [5], predict which patients may experience post-partum depression [6, 7], and estimate those who may be at risk of multiple chronic diseases including diabetes, osteoarthritis, and hypertension [8, 9]. Although a flurry of research is focused on the development of AI tools, implementation into practice remains one of the most challenging barriers to overcome.

To identify and ease these barriers, researchers have developed models related to technology implementation including the 'Technology Acceptance Model' (TAM) and the 'Unified Theory of Acceptance and the Use of Technology' (UTAUT) model [10, 11]. These are well studied models that explore the reasons that users accept or reject a given technology and how

acceptance can be improved, but unfortunately are not specific to primary healthcare or even to the healthcare at all. Another model, Sittig and Singh's sociotechnological model for studying health information technology includes eight dimensions (hardware and software computing infrastructure, clinical content, human-computer interface, people, workflow and communication, internal organizational features, external rules and regulations, and measurement and monitoring [12]. Sittig and Singh's model focuses specifically on implementation of technology in complex adaptive healthcare systems and will be used as a basis of comparison for the findings of this study.

Other researchers including Lovejoy et al. and Singh et al. have conducted reviews to explore the challenges of implementing AI in healthcare settings. The recent review by Lovejoy et al. explores the challenges of implementing AI to healthcare by breaking down the AI innovation process into three stages–invention, development, and implementation. Specifically with respect to the implementation stage, they highlight three main barriers of focus (generalizability, regulation, and deployment). Generalizability refers to the need for diversity within the training set, regulation refers to the need for ongoing monitoring of an algorithm or tool throughout its lifetime to ensure continued safety, and deployment refers to the need to minimize disruption to workflow and account for interoperability between healthcare technologies [13]. Although this review is not specific to the primary care space, it provides insights into barriers that are commonly encountered in the healthcare AI space, all of which will help inform our study. Another review by Singh et al. outlines the barriers in relation to the perspective of healthcare organizations, healthcare providers, and patients. Interestingly, from the healthcare organization perspective, they discuss the role of "company culture" as important when considering the use of embracing AI in their day-to-day practice. The provider perspective outlined concerns for the consideration of racial, ethnic, gender, and other sociodemographic characteristics within the algorithm. The perspective of patients was centered on trust, including the concern for physician-patient confidentiality if AI were to be used in their care [14]. This review provides vital insight from various stakeholder perspectives and emphasizes the importance of doing so.

To effectively implement AI in primary care settings, it is critical that these barriers are explored from multiple perspectives, as emphasized by Singh et al. above. Thus, we engaged patients, providers, and healthcare leaders in a deliberative dialogue series to gain insight into perceived challenges associated with AI. These findings will help assist those that are being called upon to make pivotal decisions regarding the future of AI in primary healthcare.

## Methods

### Setting

This study was conducted virtually within Canada, a high-income country with provincially and territorial-based public health insurance that covers all necessary hospital and physician services [15]. Both within and across the provinces and territories of Canada, primary healthcare is delivered through different models. Largely, it is funded following a fee-for-service (FFS) model, but has adopted new funding models such as the enhanced fee-for-service model and capitation model. For example, in Ontario, the enhanced FFS model is similar to the traditional FFS model, except that it includes additional targeted fee increases, extended hours, performance-based initiatives, and patient enrollment. In contrast, the capitation model reimburses primary care physicians for each person attributed to them, based on age and sex adjusted factors [16]. More recently, Ontario has been forming Ontario Health Teams including family physicians, nurse practitioners, social workers, dietitians, and other healthcare professionals to increase interdisciplinary care for a given community [17].

## Design

This emergent qualitative study used virtual deliberative dialogues. Deliberative dialogue is a participatory method in which people affected by an issue are gathered to provide advice to decisionmakers [18–20]. With facilitator support, participants work together to interpret existing evidence and identify approaches to the issue that reflect their—sometimes conflicting—values and tacit knowledge [21, 22]. We selected this method because we believe that the values and wisdom of primary care stakeholders can guide health AI in directions that support equity and advance the Quadruple Aim (optimizing health system performance through improved patient experience, improved clinical experience, better outcomes, and lower costs [21–23]).

To adhere to COVID-19 public health restrictions, we held twelve 90-minute dialogues by video conference between September 8th and October 15th, 2020. Dialogues were divided into three rounds and all sessions were facilitated by one trained Master's-level graduate student (TLU). Before taking part, participants reviewed an information package that provided an overview of health AI, key definitions, examples of AI in medicine, possible uses in primary care, and ethical challenges. The study was approved by the research ethics boards of Unity Health Toronto and the University of Toronto. Verbal informed consent was obtained from all participants and documented prior to scheduling their first dialogue session.

## Participants

Patients and primary care providers were recruited under distinct purposive variation sampling frames from various online advertisement platforms and email distribution lists. This recruitment method is outlined further by Patton et al. [24], and was chosen in order to optimize variation in age, gender, race and ethnicity, education level, income, province of residence for patients and gender, race and ethnicity, provider type, country of health professions education, years in practice, practice size, and province of practice for primary care providers. Specifically, patients were recruited using Kijiji and social media channels across 10 provinces as well as patient advisor distribution lists (patient advisory network in British Columbia, patient-family advisory council of downtown Toronto hospitals). Primary care physicians were recruited through social media channels, primary care research network news channels, and through email invitations from AP to practice colleagues and those in his network interested in digital health and health informatics. To be eligible, patients must have seen their primary care provider once within the last year and providers had to work at least one day in the clinic per week. To promote variation, we collected sociodemographic information during consent interviews and then iteratively adjusted recruitment strategies. We used a critical case frame for system leaders, seeking those involved in digital health, health informatics, or primary care governance [24]. Patients and providers were invited to take part in up to three rounds, while system leaders were invited to join one final-round session or to provide feedback on the dialogue from the first two sessions by survey. The sessions were designed this way to allow for interactive dialogue between the different perspectives of patients, providers, and health system leaders in real-time. An honorarium of $32.50 USD was offered to participants for every session attended. TLU was acquainted with two patient participants. AP was known to most providers in the study and observed three sessions.

## Data collection & analysis

Audiovisual recordings and fieldnotes were generated for each session by one of the observers (ACN, JM). Following a session, TLU and the designated observer met after each session to analyze fieldnotes using interpretive description methods, coding for concerns, perceived barriers, and facilitators of AI implementation in primary care [25–28]. This analysis guided the

emergent nature of the study (e.g., participant-facing summaries, use cases, etc.) and generated an initial code list. Early codes were later confirmed and expanded upon through dual-coder thematic analysis of verbatim transcripts (TLU, KD). Transcript analysis occurred in two passes, the first being entirely inductive and the second using agreed upon codes. TLU, KD, and AP then met to review coded transcripts and agree on final interpretations. The final interpretations were compiled into four major themes and then compared to Sittig and Singh's sociotechnical model for studying health information technology (HIT) in complex adaptive health systems [12]. Data was managed using Microsoft Excel and Word. Saturation was reached for most concepts after nine sessions.

### Rigor

Dialogue guides for rounds 2 and 3 were not pilot tested because of the study's emergent nature. These guides were iterated upon and further developed as the study progressed. Investigator triangulation was achieved through the involvement of multiple researchers in data collection and analysis [29]. Data triangulation was achieved through careful comparison of field notes, primary transcripts, researcher interpretive reflections, and participant activities throughout the study. We created multiple opportunities for participants to member-check researcher interpretations. Eighteen participants provided reflections on preliminary findings at the study's conclusion.

## Results

### Participants

Three rounds of deliberative dialogue sessions were completed with 48 participants from 8 Canadian provinces. This included 22 patients, 21 primary care providers, and 5 health system leaders; participants' characteristics can be found in S1 Table and have been published elsewhere [30].

### Findings

From these dialogue sessions, a series of barriers and strategies to implement AI in primary care emerged, which were grouped into four major themes: (1) system and data readiness, (2) the potential for bias and inequity, (3) the regulation of AI and big data, and (4) the importance of people as technology enablers. Within each of these 4 themes, participants spoke about trust, either in reference to trust in the technology or in the provider using it, so trust will be discussed within each theme. When asked what they thought about algorithm explainability, all but one of the patient participants agreed that they would trust AI algorithms used in their care by proxy of the trust they placed in their provider. Patients in one session suggested that health AI-enabled harms could damage public trust in health institutions. Patients and providers across sessions identified regulatory oversight, scientific evidence, and participatory co-development processes as builders of trust.

### System and data readiness

Significant concern around adoption and the readiness of Canada's health system came up in many sessions and was particularly focused on both the readiness of our current technology/data, as well as the willingness for clinics to engage with and adopt new AI technologies. Some primary care clinics still operate using paper and fax machines, and therefore implementing AI may be too advanced. Additionally, primary care clinics operate independently and are fragmented from one another. Each clinic "enters things in all different ways. We have to make sure that the tool is good enough to manage the many different ways that people get

information and enter information" (PR-7905). Creating an adoption strategy and designing tools to accommodate each clinic may be challenging.

Participants acknowledged the potential for AI to improve health and healthcare. However, a few patients and most providers recognized that the optimal performance of health AI tools depends upon a level of HIT interoperability that does not exist in many jurisdictions. For AI tools to reliably inform clinical decision-making and support integrated, equitable care, providers believed that algorithms must synthesize data from multiple sources, including administrative databases and EHRs across levels of care and care settings. They also believed that certain advanced AI tools (e.g., rare disease diagnosis) would have to synthesize an exponentially increasing body of evidence and ever-changing guidelines to be clinically useful.

Furthermore, when discussing the willingness to adopt AI technology, it was highlighted that gaining buy-in from clinic owners and operators may be challenging since many of them were promised a series of quality improvements during the EHR rollout that were not met. As one participant suggested, "the quality improvements that were expected didn't happen" (PR-4404). They may be hesitant to invest and place trust in an AI system unless they can clearly and quickly realize the benefit of doing so.

## Potential for bias and inequity

With varying levels of technological readiness and challenges with clinic uptake, the potential for biased algorithms and inequitable access to AI tools becomes increasingly concerning. Several providers worried that lagging digitization in resource-constrained settings (e.g., rural communities), poor interoperability of established HIT systems, or data monopolies held by leading EHR vendors could negatively impact health equity by driving the uneven application of AI tools across settings, or limiting datasets available for developing AI tools to smaller, non-representative populations, thereby increasing the risk of bias and inequity.

The potential for bias was a concern that came up in every dialogue session. Some participants brought up past studies where AI tools were proven to further perpetuate the biases of their developers and of society. As one patient participant mentioned, "there could be bias in your software. Because these factors are not all equal, they have different weights of importance. And so how that's applied will affect how good the results are going to be" (PT-5981). There was also concern for the bias that could be present in the guidelines and criteria that feed the AI algorithms. Participants were worried that if pharmaceutical companies or other private enterprises sponsored the tool, they may bias the algorithms to produce outcomes that favor their product. Participants linked this theme to notions of trust. As evidenced by the following quotation, "we want to make sure that the tools we use don't create new problems. . . they are an opportunity to address some of the biases that already exist in our system. . .and that these tools are probably only as good as the data we provide them" (PR-7905).

Equity and access to AI technology were discussed from many different perspectives throughout the dialogue sessions. From a patient perspective, there was expressed concern that some populations may not want to or be able to interact with AI tools, which as a result, may further deter them from the health system as a whole and widen the care gap that already exists. From a primary care clinic and provider level, the clinics that are frequently last to procure EHRs and new technology will likely also be the last to adopt a new AI tool; thus, we may further widen the divide between the advantaged and disadvantaged primary care practices and the individuals they treat. Overall, while participants are hopeful for the potential that AI holds, it will likely not serve populations that most frequently "fall through the cracks" (PT-4653), and therefore it is important to consider these populations and their ability to access and trust the tool before it is implemented widely in primary care.

### Regulation of AI and big data

Participants from all three groups believed that inadequate regulatory oversight of health AI tools places the public at risk of harm to physical health or wellbeing. This was a perceived barrier to health AI adoption because it decreased patient and provider trust in available tools. To feel comfortable using an AI tool in clinical decision-making, providers wanted assurance that it had been approved by a regulatory authority after meeting rigorous standards of performance, in terms of accuracy, risk of bias, and patient safety. Participants from all three groups, but especially patients, were concerned that existing privacy laws do not adequately prevent AI developers from monetizing PHI or allowing PHI to be used in ways that impact patient wellbeing (e.g., insurance or loan decisions). One provider stated that they would like to be aware of "how this data is being used and whether there were any concerns about this data being used by a [private] company" (PR-7905) and a patient mentioned, "I think it's really important to clearly set out what the data be used for and ensure that the patient understands how the confidential confidentiality and privacy still applies in those situations" (PT-4653). Evidently, all participants stressed the importance of regulatory oversight to ensure the tools are frequently monitored for safety and that there is insight into the ownership, control, and access of health data to ensure it is not being monetized or used inappropriately.

### Importance of people as technology enablers

People play a vital role in the success of any given technology; therefore, the barriers that impact the people using and affected by the technology must be properly mitigated. From the dialogue sessions, these barriers included the need for provider training and skill preservation, the fear of changing the provider-patient relationship, and the challenge of designing tools to properly suit the user's needs. Firstly, many providers were unsure about the training that would be needed to successfully adopt a given AI tool and were worried that it may impede their clinic workflow. As stated by one provider, "oftentimes, you have to adapt your workflow and sometimes your clinical processes to accommodate the tool where I really strongly feel it needs to be the reverse" (PR-4404). After being trained on the tool, some were concerned that the clinical skills and reasoning of primary care providers could be at risk due to dependence on the tool. As one provider said, "we don't want to lose that 'art of medicine' that we have learned over the years, because it's a very challenging profession where you see people with undifferentiated complaints, so you have to know something about everything. And with AI, if we rely too much on it, we might lose that clinical judgment." (PR-1783).

Secondly, some participants feared that these technologies could harm patient-provider relationships, while others thought it may enhance them. For example, one patient mentioned, "I want my doctor to be present with me in a conversation and not staring at the computer and looking at all the tools or boxes that they need to check off" (PT-4159). Many agreed with this notion, but some did refute and stated that if the AI tools are designed correctly, they may provide more space and time for relationship building and trust between the patient and the provider. Lastly, all participants agreed that the tools must be designed deliberately, to beneficially serve the needs of the users and not contribute to patient, provider, or administrative burnout.

### Strategies to promote adoption

To promote the adoption of health AI in primary care and beyond, participants offered a host of strategies to build patient and provider trust in the technology based on each theme (summarized in Table 1). Of these, the most robustly supported was a participatory AI co-development model that meaningfully involves patients, interprofessional primary care providers,

**Table 1. Strategies to promote adoption based on each theme.**

| All four themes | |
|---|---|
| *Participatory Co-design* | Ensure all stakeholders are at the table during both the design and implementation of these tools. Include representative stakeholders from different communities and different types of users (administrative clerks, providers, patients etc.) |
| *Design and Implement Iteratively* | Use iterative design and implementation strategies to allow for stakeholder feedback throughout. This will ensure the tool is designed for those that it will affect (and not the agenda of a private business) |
| *Learn from Pilot Projects* | Use pilot projects as examples to learn |
| **Theme 1: System and data readiness** | |
| *Interoperability* | Invest in interoperability to enable different technological systems to "talk to one another" thereby enabling integration between clinics and hospitals |
| *Technology Infrastructure and Architecture* | Organize technology infrastructure and architecture to support AI tools |
| *Create a Robust Adoption Strategy* | Arrange for adoption strategy where providers can slowly ease into using the technology. It may begin with tighter constraints at the start, and overtime as trust is built, it can be given more autonomy (staged approach). This strategy should also be versatile enough to be applicable to many different areas of primary care clinics. |
| **Theme 2: Potential for bias and inequity** | |
| *Unbiased Training Data* | Use unbiased, representative, good quality data to feed algorithms |
| *Prevent Bias through Frequent Re-assessment* | Ensure AI tool remains free of bias overtime by conducting routine re-assessments and analyses of the tools |
| *Ethics Training* | Consider designing ethics training courses for all of the stakeholders who will be involved in the design, testing, implementation, and use of AI tools |
| **Theme 3: Regulation of AI and big data** | |
| *Leadership Commitment* | Encourage health leaders and government players to demonstrate commitment to AI (lead by example) |
| *National Rules & Regulations* | Create and implement national AI rules and regulations for healthcare (there is a need for content standards and protocols to further enable interoperability, and a need for regulatory standards to ensure representation and bias mitigation within the algorithms and the training data sets) |
| *Design Regulations based on Level of Risk* | Create regulations based on the level of risk that the tool may pose to its users. For example, there may be different regulations for tools that conduct administrative-related functions than those that are designed for clinical decision making and diagnostic functions |
| **Theme 4: Importance of people as technology enablers** | |
| *Value Proposition* | Create a clear value proposition for each user/person affected by technology (what's in it for them? What benefits will they see from this technology?) |
| *Ease into the Technology* | Start with AI technology that can complete basic operational tasks before transitioning into the implementation of complex AI systems |
| *Transparency* | Outline all of the benefits and limitations of the technology to all affected and communicate these to the users to manage expectations |
| *AI Literacy* | Provide education and transparency into what AI tools are and how they are used in order to reduce concern of the unknown and enable all stakeholders to participate in co-development conversations |
| *Deliberate Design* | Deliberately select AI technologies to align with what we want humans to do versus what we want technology to do. There is a need to be deliberate and weigh the benefits and risks before implementing AI technologies |
| *Create a Robust Adoption Strategy* | Arrange for adoption strategy where providers can slowly ease into using the technology. It may begin with tighter constraints at the start, and overtime as trust is built, it can be given more autonomy (staged approach). This strategy should also be versatile enough to be applicable to many different areas of primary care clinics. |

clinical administrative staff, HIT specialists, and health policymakers. Participants believed that stakeholder engagement must occur upstream of piloting, ideally at the conception phase, and remain ongoing. They also specified that stakeholders must be diverse in terms of sociodemographic characteristics (e.g., race or ethnicity, gender, income), geographic location, and healthcare needs, as well as professional settings for providers (e.g., telehealth, walk-in, community health centers). Through this model, described by a patient as "designing with, rather than for," stakeholders could co-construct AI tools that address shared application priorities and promote patient-centered practices. Within the model, participants viewed processes that facilitate iterative stakeholder input (e.g., a participatory piloting program) as particularly useful for ensuring usability and workflow integration, correcting for bias, and improving algorithm performance through feedback. To specifically help overcome the barriers associated with system and data readiness, investments are required to allow for different technological systems to "talk to one another". This would thereby enable integration between primary care clinics using different EHR technologies.

The second theme, the potential for bias and inequity, may begin to be mitigated by promoting the use of unbiased, representative, good quality data to feed algorithms. It was recommended by one participant that to ensure the AI tool remains free of bias overtime, it should be re-assessed and analyzed routinely to check for bias. There was an expressed desire for ethics training courses for those involved at many stages of the design, implementation, and evaluation life cycle of the technology.

Strategies to assist with the regulation of AI and big data will require strong leadership commitment by health leaders and government players. Providers felt that this commitment to health AI would act as an example for primary care clinics to follow. Additionally, to increase trust in the technology, national AI rules and regulations for healthcare must be designed and implemented. There was an expressed need for content standards and protocols to further enable interoperability, and a need for regulatory standards to ensure representation and bias mitigation within the algorithms and training data sets.

Building trust, preventing burnout, and preserving patient-provider relationships begins with outlining a clear value proposition for each user and individual affected by the technology. It was highlighted that the benefits, limitations, and uses of the technology must be transparently communicated. Following this, simple AI technology that can complete basic operational tasks should be implemented first, before transitioning into the use of complex AI systems to encourage uptake and trust in the technology.

## Discussion

The barriers and strategies were derived from dialogue discussions conducted with patients, providers, and healthcare leaders across the country. The four major themes that arose from these discussions were in relation to system and data readiness, the potential for bias and inequity, the regulation of AI and big data, and the importance of people as technology enablers. Participatory co-development and iterative design were the two most important strategies that were heard during the sessions, and span across all four themes. This study also yielded rich insights on priority applications, which have been published elsewhere.

Sittig and Singh's sociotechnological model for studying health information technology in complex adaptive healthcare systems demonstrates the intricate challenges involved in the design, development, implementation, use, and evaluation of technology in healthcare settings [12]. It was designed to address the both the sociological and technological challenges that are present within healthcare settings that are high-pressured, fast-paced, and fragmented. Their model introduces eight interacting dimensions: hardware and software computing

infrastructure, clinical content, human-computer interface, people, workflow and communication, internal organizational features, external rules and regulations, and measurement and monitoring [12]. This model was used as a comparison of our four themes and was chosen over other models such as the Technology Acceptance Model (TAM) and the Unified Theory of Acceptance and Use of Technology (UTAUT) for its applicability to complex health systems [10, 11]. The TAM and UTAUT aim to understand why users accept or reject a given technology and how acceptance can be improved, but recent reviews show that they have failed to predict acceptance of technologies in healthcare [31]. This may be because of healthcare's unique cultural, social, and organizational factors that influence technology adoption. Therefore, Sittig and Singh's model remained the most appropriate choice for our comparison due to its specificity to health information technology.

Interestingly, the four themes captured throughout this dialogue do align with most of the eight dimensions quite remarkably. Most notably, Sittig and Singh's "people" dimension aligns with our fourth theme, the importance of people as technology enablers, as both recognize the importance of provider training and the user's perception of the technology.

Additionally, our regulation of AI and big data theme correlates with Sittig and Singh's "external rules and regulations" dimension since both are focused on the directives that must be designed at a government level to ensure all AI technology is held to a safe and reliable standard.

The system and data readiness theme does not align strongly with any one given dimension but correlates loosely with both the "hardware and software computing infrastructure" and "clinical content" dimensions. The former covers the technological perspective, including equipment and software, which could be extended to cover the contents of our theme including interoperability and digitization of medical records, while the latter addresses personal health information, data quality, and data structure.

The potential for bias and inequity theme is also not explicitly present in Sittig and Singh's eight dimensions, possibly because AI has the potential to present biases in ways that technology was less capable of before. Furthermore, trust, the overarching foundation to all of our themes is also not present in the eight dimensions but was very important to the participants during the dialogue sessions.

Overall, Sittig and Singh's sociotechnological framework is advantageous to consider both the social and technical perspectives of technology implementation projects, but based on our deliberative dialogue series, we believe that new dimensions (*e.g.*, trust, bias, and equity) would be required to translate this model to be applicable to AI implementation projects in the primary care space now and in the future.As mentioned in the introduction, the recent review conducted by Lovejoy et al. divides the AI innovation process into three stages (invention, development, and implementation) and outlines key considerations for innovators during each phase [13]. The considerations of the third stage, implementation, include generalizability, regulation, and deployment. Generalizability refers to the need for diversity within the training set, regulation refers to the need for ongoing monitoring of an algorithm or tool throughout its lifetime to ensure continued safety, and deployment refers to the need to minimize disruption to workflow and account for interoperability between healthcare technologies [13]. When comparing the four themes uncovered in our study to the implementation stage of this review, it is evident that our deliberative dialogue results greatly support those of Lovejoy et al. For example, within our 'potential for bias and inequity' and 'regulation of AI and big data' themes, we heard multiple times in our sessions that diversity within training sets and continuous monitoring of a given tool are both vital to ensure bias and safety are maintained over time. This is very similar to Lovejoy et al.'s generalizability and regulation sections as they state, "The training data must be at least as diverse as the population that the algorithm intends

to serve" and "Good performance at the time of deployment does not guarantee that the model will continue to perform well. This introduces the need to regulate throughout the life-time of an algorithm, and the need to continually demonstrate safe and effective practices" [13]. Therefore, our 'potential for bias and inequity' theme is similar to Lovejoy et al.'s concept of generalizability, and our 'regulation of AI and big data' compliments the regulation section explained in their review.

The review by Singh et al. outlines AI implementation barriers in relation to the perspective of healthcare organizations, healthcare providers, and patients. These three perspectives are similar to those included in our study, except that instead of healthcare organizations we heard from health system leaders. From the review's healthcare organization, they discussed the role of "company culture" as important when considering the use of embracing AI in their day-to-day practice–a notion that did not come up explicitly in our deliberative dialogue sessions. The provider perspective outlined concerns for the consideration of racial, ethnic, gender, and other sociodemographic characteristics within the algorithm. This was similar to what was heard in our 'Potential for bias and inequity' theme, except that we heard the concern from all three types of stakeholders and not just from the providers alone. The perspective of patients was centered on trust, including the concern for physician-patient confidentiality if AI were to be used in their care [14]. This compliments our finding of the importance of trust, however, in our sessions it was woven into all four themes and from all perspectives (patients, providers and health system leaders).

The strengths of this study include the use of a two-coder analysis approach, its transferability to non-Canadian settings with similar primary care models, and the use of informed inter-professional stakeholder engagement. First, for the analysis of the data, a traditional two-coder approach was employed to ensure trustworthiness and intercoder reliability. Second, based on the services and values that are offered by most primary care clinics [32], our findings are applicable to jurisdictions outside of Canada, confirming interjurisdictional transferability. Last, participants in these sessions were from many different backgrounds (patients, primary care providers, and system leaders), but all learned about AI technology before taking part, nurturing an informed discussion. The interprofessional design of these sessions enabled participants to refute and agree on discussion topics and fostered the ability to explore different perspectives of using AI in primary care.

To further interpret this study, it is essential to consider the limitations. It was completed by engaging participants from eight Canadian provinces but unfortunately did not include the perspectives from Canada's territories or any self-identifying Indigenous people, both of whom may have unique perspectives regarding barriers to using AI in primary healthcare [33]. To further improve this study and its transferability, future work should focus on the perspectives of those that self-identify as indigenous as well as individuals form Canada's territories. This would enable more diverse perspectives and would be vital before any AI implementation projects are considered. Additionally, due to the ongoing COVID-19 pandemic, we were only able to include five health system leaders in the study. The health leaders that were able to attend provided rich system-level perspectives that were unique to those heard from patients and providers, thus including more system leaders would have been preferred.

This level of study (48 participants from 8 Canadian provinces including 22 patients, 21 primary care providers, and 5 health system leaders) provides an excellent starting point for the barriers and strategies to use when implementing AI in primary care settings, but a study with larger reach would be recommended before practical AI applications become commonplace in primary care. To increase reach and gain further insight, a deliberative dialogue would not be practical but instead a survey could gain the perspectives of more of the population. This in addition to a focused dialogue with individuals from Canada's territories and those who self-

identify as indigenous would be vital before practical AI solutions are implemented widely in primary care.

## Conclusions

Participants were hopeful for AI technology, but perceived barriers to its impact, most of which converged on trust in the technology, and thereby patient and provider willingness to adopt. These barriers were consolidated into four key themes and strategies to overcome these barriers were proposed, including most notably, participatory co-design and iterative implementation. These findings will act as a critical catalyst to address these barriers, as they highlight patient, provider and health system leaders concerns regarding implementation of AI in primary care settings.

## Supporting information

**S1 Checklist. COREQ.**
(PDF)

**S1 Table. Study participants.**
(DOCX)

**S2 Table. Iterative coding process.**
(PDF)

**S1 Appendix. Participant AI learning module.**
(DOCX)

## Acknowledgments

We are grateful to the study participants who made this research possible by sharing their time and rich perspectives. We thank Millie Upshaw, Sally Headrick, Monica Brands, and Jane Cooney for reviewing the online informational module used to support participants.

## Author Contributions

**Conceptualization:** Tara Upshaw, Jennifer Gibson, Andrew D. Pinto.

**Data curation:** Tara Upshaw.

**Formal analysis:** Katrina Darcel, Tara Upshaw, Timothy C. Y. Chan.

**Funding acquisition:** Andrew D. Pinto.

**Investigation:** Tara Upshaw, Amy Craig-Neil, Jillian Macklin, Andrew D. Pinto.

**Methodology:** Andrew D. Pinto.

**Project administration:** Amy Craig-Neil, Andrew D. Pinto.

**Resources:** Andrew D. Pinto.

**Supervision:** Carolyn Steele Gray, Timothy C. Y. Chan, Jennifer Gibson, Andrew D. Pinto.

**Writing – original draft:** Katrina Darcel.

**Writing – review & editing:** Katrina Darcel, Tara Upshaw, Amy Craig-Neil, Jillian Macklin, Carolyn Steele Gray, Timothy C. Y. Chan, Jennifer Gibson, Andrew D. Pinto.

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
