## [Decision Letter · Decision Letter 0]

6 Sep 2022

PONE-D-22-20397Implementing artificial intelligence in Canadian primary care: barriers and strategies identified through a national deliberative dialoguePLOS ONE

Dear Dr. Pinto,

Thank you for submitting your manuscript to PLOS ONE. After careful consideration, we feel that it has merit but does not fully meet PLOS ONE’s publication criteria as it currently stands. Therefore, we invite you to submit a revised version of the manuscript that addresses the points raised during the review process.

We look forward to receiving your revised manuscript.

Kind regards,

Ardashir Mohammadzadeh, Phd

Academic Editor

PLOS ONE

Journal Requirements:

2.We note that the grant information you provided in the ‘Funding Information’ and ‘Financial Disclosure’ sections do not match. 

Additional Editor Comments:

Revise the paper according to the comments of reviewers; add some comparisions; add some statements and references about the potential improvement by type-3 fuzzy logic systems;

Reviewers' comments:

Reviewer's Responses to Questions

**Comments to the Author**

1. Is the manuscript technically sound, and do the data support the conclusions?

Reviewer #1: Yes

Reviewer #2: Yes

2. Has the statistical analysis been performed appropriately and rigorously? 

Reviewer #1: Yes

Reviewer #2: N/A

3. Have the authors made all data underlying the findings in their manuscript fully available?

Reviewer #1: Yes

Reviewer #2: Yes

4. Is the manuscript presented in an intelligible fashion and written in standard English?

Reviewer #1: Yes

Reviewer #2: Yes

5. Review Comments to the Author

Reviewer #1: Overall the article is impressive and it highlights insights into practicalities of AI integration in primary care. It shows good practice by adhering to COREQ reporting guidelines for qualitative work, and it makes good use of quotes and descriptors to illustrate the themes. The work is novel and the conclusions will have an impact in the field. I would recommend accepting for publication but have a few suggestions for ways to add value to the current work:

1. There needs to be more examples of existing uses of AI in the introduction to put the current work in better context. In relation to this, any examples of AI application in primary care that were discussed in the dialogues should be expanded on if it would help explain the existing comments and quotes by participants.

2. I note with interest the sections mentioning the limitation of processing longitudinal clinical data for AI use, and think this is worth expanding on as a significant practical difficulty.

3. It would benefit your international audience to explain more about the health system in Canada as well as the terminology you have used to describe it e.g. what is a team-based care model?

4. What is the quadruple aim and support equity?

5. Please provide the participant information sheets handed out before the dialogues as supplemental data

6. Please list which advertisement platforms were used and acknowledge any patient groups that were used.

7. More details on sampling method would be beneficial

8. Line 227 "publics" should be "populations"

9. It would be useful to write provide a further description of sittig and sing's sociotechnological model which was used as a framework for analysis. In particular, why was it chosen in lieu of other prevailing models such as the unified theory of acceptance and use of technology (UTAUT), and technology acceptance model (TAM)?

10. If there are any coding tables available to visualise how the final themes were reached from the primary codes, these would be useful to put into supplementary data so that the iterative process can be demonstrated.

Reviewer #2: This is a well written article on an important topic. The main barrier to publication, in my view, is the weakness of the literature review.

There are other papers which have explored barriers to implementation, for example: https://doi.org/10.7861/fhj.2021-0128 and https://www.ncbi.nlm.nih.gov/pmc/articles/PMC7443115/

This paper is good opportunity to highlight challenges raised by existing literature with empirical evidence.

The Introduction section is very concise and superficial at the moment. Please consider expanding it with more detailed overview of existing literature. The authors write: "There is a paucity of research that focuses specifically on AI implementation challenges and the strategies to overcome them within the primary care space." but they do not really mention any of the existing research in the field. There are only four references in the introduction and all of these are related to generic AI papers.

I think that with an expanded literature review in the discussion and an enhanced introduction, this paper would be a good addition to the body of literature on AI implementation science.

6. PLOS authors have the option to publish the peer review history of their article (what does this mean?). If published, this will include your full peer review and any attached files.

Reviewer #1: **Yes: **Ahmed Al-Naher

Reviewer #2: No

---

## [Author Response · Author response to Decision Letter 0]

7 Nov 2022

The attached file, Response to Reviewers, contains a point-by-point response to the specific reviewer comments table format. We also note in this document that the general comments from the Editor have been addressed: this feedback was greatly appreciated, and is incorporated throughout the manuscript. Below we have included the text of our response, however we would suggest using the attachment to more clearly understand the changes.

--

Dear PLoS ONE Editorial Team,

We appreciate the time taken by our reviewers and the editorial team to review our manuscript entitled, Implementing artificial intelligence in Canadian primary care: barriers and strategies identified through a national deliberative dialogue. We appreciate the opportunity to resubmit this paper, and we have addressed each concern raised. 

We look forward to hearing from you about this manuscript.

--

Reviewer #1:

1. There needs to be more examples of existing uses of AI in the introduction to put the current work in better context. In relation to this, any examples of AI application in primary care that were discussed in the dialogues should be expanded on if it would help explain the existing comments and quotes by participants. • Examples added: “Examples of risk prediction AI tools applicable to the primary care space include machine learning algorithms that identify primary care patients at risk for cancer (3), predict which patients may experience post-partum depression (4,5), and estimate those who may be at risk of multiple chronic diseases including diabetes, osteoarthritis, and hypertension (6,7). Although a flurry of research is focused on the development of AI tools, implementation into practice remains one of the most challenging barriers to overcome.”

• Applications discussed in the dialogue were theoretical cases to provide a basis for discussion.

• Also added more literature review on AI implementation challenges in health care:

“A recent review by Lovejoy et al. explores these challenges to health care as a whole by breaking down the AI innovation process into three stages – invention, development, and implementation. Specifically with respect to the implementation stage, they highlight three main areas of focus (generalizability, regulation, and deployment). Generalizability refers to the need for diversity within the training set, regulation refers to the need for ongoing monitoring of an algorithm or tool throughout its lifetime to ensure continued safety, and deployment refers to the need to minimize disruption to workflow and account for interoperability between health care technologies. Although this review is not specific to the primary care space, it provides insight into some of the barriers to AI implementation that exist within the health care space. Another review by Singh et al. outlines the barriers in relation to the perspective of health care organizations, health care providers, and patients. Interestingly, from the health care organization perspective, they discuss the role of “company culture” as important when considering the use of embracing AI in their day-to-day practice. The provider perspective outlined concerns for the consideration of racial, ethnic, gender, and other sociodemographic characteristics within the algorithm. Finally, the patients perspective was based on trust, and what physician-patient confidentiality would look like when AI is used in their care. Although these reviews are not specific to the primary care space, they provide insight into some of the barriers to AI implementation that exist within the health care space.”

2. I note with interest the sections mentioning the limitation of processing longitudinal clinical data for AI use, and think this is worth expanding on as a significant practical difficulty. • Just to clarify, the introduction notes that longitudinal clinical data is one of the reasons that is unique to primary care data, but not necessarily a limitation.

3. It would benefit your international audience to explain more about the health system in Canada as well as the terminology you have used to describe it e.g. what is a team-based care model? • Re-wrote the “Setting” section to be more comprehensive of the models used in Canada, and to explain that healthcare is provincially/territorially managed:

“This study was conducted virtually within Canada, a high-income country with provincially and territorial-based public health insurance that covers all necessary hospital and physician services (5). Both within and across provinces and territories of Canada, primary health care is delivered through different models. Largely, it is funded following a fee-for-service (FFS) model, but has adopted new funding models such as the enhanced fee-for-service model and capitation model. In Ontario, the enhanced FFS model is similar to the traditional FFS model, except that it includes additional targeted fee increases, extended hours, performance-based initiatives, and patient enrollment. In contrast, the capitation model reimburses primary care physicians for each person attributed to them, based on age and sex adjusted factors (6). More recently, Ontario has been forming Ontario Health Teams including family physicians, nurse practitioners, social workers, dietitians, and other health care professionals to increase interdisciplinary care for a given community (7).”

4. What is the quadruple aim and support equity? • Supporting equity put before quadruple aim so it did not sound like it was all one thing. 

• Added definition within text to clarify: “optimizing health system performance through improved patient experience, improved clinical experience, better outcomes, and lower costs”

5. Please provide the participant information sheets handed out before the dialogues as supplemental data • AI learning module will be attached when revisions are re-uploaded. It is named “Supplemental Material – Participant AI Learning Module”

6. Please list which advertisement platforms were used and acknowledge any patient groups that were used. • Added this to the manuscript under Participants section:

“Specifically, patients were recruited using Kijiji and social media channels across 10 provinces as well as patient advisor distribution lists (patient advisory network in British Columbia, patient-family advisory council of downtown Toronto hospitals). Primary care physicians were recruited through social media channels, primary care research network news channels, and through email invitations from AP to practice colleagues and those in his network interested in digital health and health informatics.”

7. More details on sampling method would be beneficial • Added more detail for sampling method: “This recruitment method is outlined further by Patton et al (21), and was chosen in order to optimize variation in age, gender, race and ethnicity, education level, income, province of residence for patients and gender, race and ethnicity, provider type, country of health professions education, years in practice, practice size, and province of practice for primary care providers.”

8. Line 227 "publics" should be "populations" • Switched from “publics” to “populations”

9. It would be useful to write provide a further description of sittig and sing's sociotechnological model which was used as a framework for analysis. In particular, why was it chosen in lieu of other prevailing models such as the unified theory of acceptance and use of technology (UTAUT), and technology acceptance model (TAM)? • Revised intro of model: “Sittig and Singh’s sociotechnological model for studying health information technology in complex adaptive healthcare systems demonstrates the complex challenges involved in the design, development, implementation, use, and evaluation of technology in health care settings. It was designed to address the both the sociological and technological challenges that are present within health care settings that are high-pressured, fast-paced and distributed.”

• Added explanation: “This model was used as a comparison of our four themes and was chosen over other models such as the Technology Acceptance Model (TAM) and the Unified Theory of Acceptance and Use of Technology (UTAUT) for its applicability to health care. The TAM and UTAUT aim to understand why users accept or reject a given technology and how acceptance can be improved, but recent reviews show that they have failed to predict acceptance of technologies in health care (23). This may be because of health care’s unique cultural, social, and organizational factors that influence technology adoption. Therefore, Sittig and Singh’s model remained the most appropriate choice for our comparison due to its specificity to health information technology.”

10. If there are any coding tables available to visualise how the final themes were reached from the primary codes, these would be useful to put into supplementary data so that the iterative process can be demonstrated. • Coding table attached to re-submission

Reviewer #2:

This is a well written article on an important topic. The main barrier to publication, in my view, is the weakness of the literature review. There are other papers which have explored barriers to implementation, for example: https://doi.org/10.7861/fhj.2021-0128 and https://www.ncbi.nlm.nih.gov/pmc/articles/PMC7443115/ This paper is good opportunity to highlight challenges raised by existing literature with empirical evidence. The Introduction section is very concise and superficial at the moment. Please consider expanding it with more detailed overview of existing literature. The authors write: "There is a paucity of research that focuses specifically on AI implementation challenges and the strategies to overcome them within the primary care space." but they do not really mention any of the existing research in the field. There are only four references in the introduction and all of these are related to generic AI papers. I think that with an expanded literature review in the discussion and an enhanced introduction, this paper would be a good addition to the body of literature on AI implementation science. • Added findings by one of the reccomended papers by Reviewer #2: “A recent review by Lovejoy et al. explores these challenges to health care as a whole by breaking down the AI innovation process into three stages – invention, development, and implementation. Specifically with respect to the implementation stage, they highlight three main areas of focus (generalizability, regulation, and deployment). Generalizability refers to the need for diversity within the training set, regulation refers to the need for ongoing monitoring of an algorithm or tool throughout its lifetime to ensure continued safety, and deployment refers to the need to minimize disruption to workflow and account for interoperability between health care technologies. Although this review is not specific to the primary care space, it provides insight into some of the barriers to AI implementation that exist within the health care space.”

• Also from the second recommended manuscript from reviewer #2: “Another review by Singh et al. outlines the barriers in relation to the perspective of health care organizations, health care providers, and patients. Interestingly, from the health care organization perspective, they discuss the role of “company culture” as important when considering the use of embracing AI in their day-to-day practice. The provider perspective outlined concerns for the consideration of racial, ethnic, gender, and other sociodemographic characteristics within the algorithm. Finally, the patients perspective was based on trust, and what physician-patient confidentiality would look like when AI is used in their care. Although these reviews are not specific to the primary care space, they provide insight into some of the barriers to AI implementation that exist within the health care space.”

---

## [Decision Letter · Decision Letter 1]

2 Dec 2022

PONE-D-22-20397R1Implementing artificial intelligence in Canadian primary care: barriers and strategies identified through a national deliberative dialoguePLOS ONE

Dear Dr. Pinto,

Thank you for submitting your manuscript to PLOS ONE. After careful consideration, we feel that it has merit but does not fully meet PLOS ONE’s publication criteria as it currently stands. Therefore, we invite you to submit a revised version of the manuscript that addresses the points raised during the review process.

We look forward to receiving your revised manuscript.

Kind regards,

Ardashir Mohammadzadeh, Phd

Academic Editor

PLOS ONE

Reviewers' comments:

Reviewer's Responses to Questions

**Comments to the Author**

1. If the authors have adequately addressed your comments raised in a previous round of review and you feel that this manuscript is now acceptable for publication, you may indicate that here to bypass the “Comments to the Author” section, enter your conflict of interest statement in the “Confidential to Editor” section, and submit your "Accept" recommendation.

Reviewer #2: (No Response)

Reviewer #3: (No Response)

2. Is the manuscript technically sound, and do the data support the conclusions?

Reviewer #2: Yes

Reviewer #3: Yes

3. Has the statistical analysis been performed appropriately and rigorously? 

Reviewer #2: N/A

Reviewer #3: Yes

4. Have the authors made all data underlying the findings in their manuscript fully available?

Reviewer #2: Yes

Reviewer #3: Yes

5. Is the manuscript presented in an intelligible fashion and written in standard English?

Reviewer #2: Yes

Reviewer #3: Yes

6. Review Comments to the Author

Reviewer #2: My original review focussed on two deficiencies: the brevity of the Introduction and of the Discussion. The authors have adequately expanded the Introduction and I hope they agree that this has strengthened the manuscript. However, the lack of comparison to existing literature in the Discussion remains a weakness of the manuscript. The Discussion section of the manuscript remains sparse and there is only one other relevant paper discussed: "Sittig and Singh’s sociotechnological model for studying health information technology". It would be advisable to research previous studies using surveys and or other similar methodologies to identify barriers to limitations.

Reviewer #3: The paper is sufficiently innovative but needs some improvement as follows to be ready for publication in the journal:

1- The comparison with other suggested or conventional methods and their discussion is not well-highlighted in the introduction. It can helps to demonstrate the strength of the work and its weakness practically.

2-Neural network and fuzzy networks are very efficient intelligent methods. How can the offered method be improved by using fuzzy systems and neural networks? It is strongly recommended to include the following paper in the introduction refer and explain .

An interval type-3 fuzzy system and a new online fractional-order learning algorithm: theory and practice.

3-"Only five health system leaders were included in the study and no self-identifying Indigenous people". According to the mentioned limitation, what is the suitable practical solution applicable for the future works improvement?

4-Is this level of study of the mentioned community effective for practical implementations? Please explain?

7. PLOS authors have the option to publish the peer review history of their article (what does this mean?). If published, this will include your full peer review and any attached files.

Reviewer #2: No

Reviewer #3: No

---

## [Author Response · Author response to Decision Letter 1]

20 Jan 2023

We appreciate the time taken by the reviewers and the editorial team to review our manuscript entitled, "Implementing artificial intelligence in Canadian primary care: barriers and strategies identified through a national deliberative dialogue." We appreciate the opportunity to resubmit this paper, and we have addressed each concern raised. We have uploaded a response letter and point-by-point response. The content of the latter is copied below, however we would suggest referring to the table for ease of interpretation.

[Note: there was no Reviewer #1 in the Decision letter.]

Reviewer #2:

My original review focussed on two deficiencies: the brevity of the Introduction and of the Discussion. The authors have adequately expanded the Introduction and I hope they agree that this has strengthened the manuscript. However, the lack of comparison to existing literature in the Discussion remains a weakness of the manuscript. The Discussion section of the manuscript remains sparse and there is only one other relevant paper discussed: "Sittig and Singh’s sociotechnological model for studying health information technology". It would be advisable to research previous studies using surveys and or other similar methodologies to identify barriers to limitations.

RESPONSE:

Thank you for the guidance on the introduction. We agree that the changes to the introduction have improved the paper greatly.

• Some changes to the discussion were made in the last round of edits (introducing the TAM and UTAUT models and why Sittig & Singh was selected over them)

• Changes to the discussion during this round of edits includes a more in-depth comparison to the Sittig & Singh model:

“The system and data readiness theme does not align strongly with any one given dimension but correlates loosely with both the “hardware and software computing infrastructure” and “clinical content” dimensions. The former covers the technological perspective, including equipment and software, which could be extended to cover the contents of our theme including interoperability and digitization of medical records, while the latter addresses personal health information, data quality, and data structure.

The potential for bias and inequity theme is also not explicitly present in Sittig and Singh’s eight dimensions, possibly because AI has the potential to present biases in ways that technology was less capable of before. Furthermore, trust, the overarching foundation to all of our themes is also not present in the eight dimensions but was very important to the participants during the dialogue sessions. 

Overall, Sittig and Singh’s sociotechnological framework is advantageous to consider both the social and technical perspectives of technology implementation projects, but based on our deliberative dialogue series, we believe that new dimensions (e.g., trust, bias, and equity) would be required to translate this model to be applicable to AI implementation projects in the primary care space now and in the future.”

• Further research was completed to compare our finding to two other reviews on barriers to implementation: 

“As mentioned in the introduction, the recent review conducted by Lovejoy et al. divides the AI innovation process into three stages (invention, development, and implementation) and outlines key considerations for innovators during each phase (13). The considerations of the third stage, implementation, include generalizability, regulation, and deployment. Generalizability refers to the need for diversity within the training set, regulation refers to the need for ongoing monitoring of an algorithm or tool throughout its lifetime to ensure continued safety, and deployment refers to the need to minimize disruption to workflow and account for interoperability between healthcare technologies (13). When comparing the four themes uncovered in our study to the implementation stage of this review, it is evident that our deliberative dialogue results greatly support those of Lovejoy et al. For example, within our ‘potential for bias and inequity’ and ‘regulation of AI and big data’ themes, we heard multiple times in our sessions that diversity within training sets and continuous monitoring of a given tool are both vital to ensure bias and safety are maintained over time. This is very similar to Lovejoy et al.’s generalizability and regulation sections as they state, “The training data must be at least as diverse as the population that the algorithm intends to serve” and “Good performance at the time of deployment does not guarantee that the model will continue to perform well. This introduces the need to regulate throughout the lifetime of an algorithm, and the need to continually demonstrate safe and effective practices” (13). Therefore, our ‘potential for bias and inequity’ theme is similar to Lovejoy et al.’s concept of generalizability, and our ‘regulation of AI and big data’ compliments the regulation section explained in their review. 

The review by Singh et al. outlines AI implementation barriers in relation to the perspective of healthcare organizations, healthcare providers, and patients. These three perspectives are similar to those included in our study, except that instead of healthcare organizations we heard from health system leaders. From the review’s healthcare organization, they discussed the role of “company culture” as important when considering the use of embracing AI in their day-to-day practice – a notion that did not come up explicitly in our deliberative dialogue sessions. The provider perspective outlined concerns for the consideration of racial, ethnic, gender, and other sociodemographic characteristics within the algorithm. This was similar to what was heard in our ‘Potential for bias and inequity’ theme, except that we heard the concern from all three types of stakeholders and not just from the providers alone. The perspective of patients was centered on trust, including the concern for physician-patient confidentiality if AI were to be used in their care (14). This compliments our finding of the importance of trust, however, in our sessions it was woven into all four themes and from all perspectives (patients, providers and health system leaders).”

• Please see response to reviewer #3 below for additional changes made to the discussion section.

REVIEWER #3 - 1/4

The comparison with other suggested or conventional methods and their discussion is not well-highlighted in the introduction. It can helps to demonstrate the strength of the work and its weakness practically.

RESPONSE:

Introduced the common tech implementation models (TAM and UTUAT). Also introduced the Sittig and Singh model in the introduction. These three models are referred to in the discussion so we agree, it is important to introduce them first in the introduction. 

“To identify and ease these barriers, researchers have developed models related to technology implementation including the ‘Technology Acceptance Model’ (TAM) and the ‘Unified Theory of Acceptance and the Use of Technology’ (UTAUT) model (10,11). These are well studied models that explore the reasons that users accept or reject a given technology and how acceptance can be improved, but unfortunately are not specific to primary healthcare or even to the healthcare at all. Another model, Sittig and Singh’s sociotechnological model for studying health information technology includes eight dimensions (hardware and software computing infrastructure, clinical content, human-computer interface, people, workflow and communication, internal organizational features, external rules and regulations, and measurement and monitoring (12). Sittig and Singh’s model focuses specifically on implementation of technology in complex adaptive healthcare systems and will be used as a basis of comparison for the findings of this study.”

Additionally, the mention of the two reviews in the introduction (Lovejoy et al. and Singh et al.) were better highlighted and explained in relation to this paper.

“Other researchers including Lovejoy et al. and Singh et al. have conducted reviews to explore the challenges of implementing AI in healthcare settings. The recent review by Lovejoy et al. explores the challenges of implementing AI to healthcare by breaking down the AI innovation process into three stages – invention, development, and implementation. Specifically with respect to the implementation stage, they highlight three main barriers of focus (generalizability, regulation, and deployment). Generalizability refers to the need for diversity within the training set, regulation refers to the need for ongoing monitoring of an algorithm or tool throughout its lifetime to ensure continued safety, and deployment refers to the need to minimize disruption to workflow and account for interoperability between healthcare technologies (13). Although this review is not specific to the primary care space, it provides insights into barriers that are commonly encountered in the healthcare AI space, all of which will help inform our study. Another review by Singh et al. outlines the barriers in relation to the perspective of healthcare organizations, healthcare providers, and patients. Interestingly, from the healthcare organization perspective, they discuss the role of “company culture” as important when considering the use of embracing AI in their day-to-day practice. The provider perspective outlined concerns for the consideration of racial, ethnic, gender, and other sociodemographic characteristics within the algorithm. The perspective of patients was centered on trust, including the concern for physician-patient confidentiality if AI were to be used in their care (14). This review provides vital insight from various stakeholder perspectives and emphasizes the importance of doing so.”

Reviewer #3 - 2/4

Neural network and fuzzy networks are very efficient intelligent methods. How can the offered method be improved by using fuzzy systems and neural networks? It is strongly recommended to include the following paper in the introduction refer and explain.

An interval type-3 fuzzy system and a new online fractional-order learning algorithm: theory and practice.

RESPONSE:

We appreciate this recommendation. We have given it careful consideration, and believe the suggested citation and related recommendation for revision are beyond the scope of the work presented in this manuscript.

Reviewer #3 - 3/4

"Only five health system leaders were included in the study and no self-identifying Indigenous people". According to the mentioned limitation, what is the suitable practical solution applicable for the future works improvement?

RESPONSE:

Added a practical solution for future work: 

“To further improve this study and its transferability, future work should focus on the perspectives of those that self-identify as indigenous as well as individuals form Canada’s territories. This would enable more diverse perspectives and would be vital before any AI implementation projects are considered.”

Reviewer #3 - 4/4

Is this level of study of the mentioned community effective for practical implementations? Please explain?

RESPONSE:

Expanded on this as follows: 

“This level of study (48 participants from 8 Canadian provinces including 22 patients, 21 primary care providers, and 5 health system leaders) provides an excellent starting point for the barriers and strategies to use when implementing AI in primary care settings, but a study with larger reach would be recommended before practical AI applications become commonplace in primary care. To increase reach and gain further insight, a deliberative dialogue would not be practical but instead a survey could gain the perspectives of more of the population. This in addition to a focused dialogue with individuals from Canada’s territories and those who self-identify as indigenous would be vital before practical AI solutions are implemented widely in primary care.”

---

## [Decision Letter · Decision Letter 2]

31 Jan 2023

Implementing artificial intelligence in Canadian primary care: barriers and strategies identified through a national deliberative dialogue

PONE-D-22-20397R2

Dear Dr. Pinto,

We’re pleased to inform you that your manuscript has been judged scientifically suitable for publication and will be formally accepted for publication once it meets all outstanding technical requirements.

Kind regards,

Ardashir Mohammadzadeh, Phd

Academic Editor

PLOS ONE

Additional Editor Comments (optional):

Reviewers' comments:

Reviewer's Responses to Questions

**Comments to the Author**

1. If the authors have adequately addressed your comments raised in a previous round of review and you feel that this manuscript is now acceptable for publication, you may indicate that here to bypass the “Comments to the Author” section, enter your conflict of interest statement in the “Confidential to Editor” section, and submit your "Accept" recommendation.

Reviewer #2: All comments have been addressed

Reviewer #3: All comments have been addressed

2. Is the manuscript technically sound, and do the data support the conclusions?

Reviewer #2: Yes

Reviewer #3: Partly

3. Has the statistical analysis been performed appropriately and rigorously? 

Reviewer #2: N/A

Reviewer #3: I Don't Know

4. Have the authors made all data underlying the findings in their manuscript fully available?

Reviewer #2: Yes

Reviewer #3: Yes

5. Is the manuscript presented in an intelligible fashion and written in standard English?

Reviewer #2: Yes

Reviewer #3: Yes

6. Review Comments to the Author

Reviewer #2: No further concerns. Comments from previous reviews have been answered. Manuscript appears much improved from original submission.

Reviewer #3: (No Response)

7. PLOS authors have the option to publish the peer review history of their article (what does this mean?). If published, this will include your full peer review and any attached files.

Reviewer #2: No

Reviewer #3: No

---

## [Editor Report · Acceptance letter]

17 Feb 2023

PONE-D-22-20397R2 

Implementing artificial intelligence in Canadian primary care: barriers and strategies identified through a national deliberative dialogue 

Dear Dr. Pinto:

I'm pleased to inform you that your manuscript has been deemed suitable for publication in PLOS ONE. Congratulations! Your manuscript is now with our production department. 

Kind regards, 

on behalf of

Dr. Ardashir Mohammadzadeh 

Academic Editor

PLOS ONE